# Experiments with Active-Set LP Algorithms Allowing Basis Deficiency

**Pablo Guerrero-García** [1],[*] and **Eligius M. T. Hendrix** [2]

1   Applied Mathematics, Universidad de Málaga, 29071 Málaga, Spain
2   Computer Architecture, Universidad de Málaga, 29071 Málaga, Spain
*   Correspondence: pablito@ctima.uma.es

**Abstract:** An interesting question for linear programming (LP) algorithms is how to deal with solutions in which the number of nonzero variables is less than the number of rows of the matrix in standard form. An approach is that of basis deficiency-allowing (BDA) simplex variations, which work with a subset of independent columns of the coefficient matrix in standard form, wherein the basis is not necessarily represented by a square matrix. We describe one such algorithm with several variants. The research question deals with studying the computational behaviour by using small, extreme cases. For these instances, we must wonder which parameter setting or variants are more appropriate. We compare the setting of two nonsimplex active-set methods with Holmström's *TomLab* LPSIMPLEX v3.0 commercial sparse primal simplex commercial implementation. All of them update a sparse QR factorization in MATLAB. The first two implementations require fewer iterations and provide better solution quality and running time.

**Keywords:** linear programming; phase I; basis deficiency-allowing simplex variations; nonsimplex active-set method; Farkas lemma; sparse matrices

## 1. Introduction

Since the introduction of the simplex method for linear programming (LP) in 1947, there has been interest in viewing algorithms from a nonlinear optimisation point of view. Most well known and developed are the ideas of interior point methods, established at the time of the first barrier methods of Dikin in 1967. Although the questions on the simplex method originate from its beginning in 1947, there is still a lot of ongoing research; in the last three years one can find publications such as [1–8], just to cite a few. Various aspects of the method are investigated in these works, where Im and Wolkowicz [6] specifically also focus on degeneracy and related issues. However, these recent works neither work with a nonsquare basis, nor do they include a sparse linear algebra procedure to efficiently solve the sparse least squares subproblems that we need in our algorithmic approach.

A generalization of the well-known dual simplex method can be found in [9,10], which discuss relaxing operations toward nonsimplex steps. One can find more recent considerations in this line in [11,12] with the incorporation of steepest-edge pivoting rules (see also [13,14]). Gill and Murray [9] presented a nonsimplex active-set algorithm for solving a problem $(P)$ starting with a feasible point $x_0$. Descriptions and analyses can also be found in [14–16]. We investigated and developed a nonsimplex active-set (NSA) method for a dual linear program in standard form $(D)$, [17]. A challenge for the development is to have to deal with high degeneracy, which for a simplex-like method provides basis-deficiency, i.e., the dual variable set has less than $n$ variables. Therefore, we aimed for the development of a method, which is also called a basis deficiency-allowing (BDA) simplex variation (see [11,12,18]).

This paper presents a study of the computational behavior of a MATLAB implementation on a set of extreme cases and sparse inequality-constrained linear programs customized

from the literature. We compare two nonsimplex active-set methods with the commercial solver of Holmström called TomLab LPSIMPLEX V3.0 sparse primal simplex implementation. Various numerical linear algebra procedures are used in MATLAB to keep a sparse QR factorization of the tall-and-skinny basis matrix updated. To elaborate upon the experiments, we introduce the notation and the main algorithm and its implementation in Section 2. We report on computational findings in Section 3. Section 4 summarises our findings.

## 2. Materials and Methods

Consider the nonsymmetric primal dual relation in linear programming, wherein we deviate from the usual notation here in exchanging $b$ and $c$, $x$ and $y$, $n$ and $m$, and $(P)$ and $(D)$ in the notation of e.g., [14]. Now consider

$$
(P) \quad \begin{array}{ll} \min & \ell(x) := c^T x \quad , \ x \in \mathbb{R}^n \\ \text{s.t.} & A^T x \geq b \end{array} \qquad (D) \quad \begin{array}{ll} \max & \mathcal{L}(y) := b^T y \quad , \ y \in \mathbb{R}^m \\ \text{s.t.} & Ay = c \qquad \quad , \ y \geq 0 \end{array} , \tag{1}
$$

where $A \in \mathbb{R}^{n \times m}$ with $m \geq n$ and $\mathrm{rank}(A) = n$. Let $\mathcal{F}$ and $\mathcal{G}$ denote the feasible region of $(P)$ and $(D)$, respectively. Here, $c$ and $b$ are, respectively, the primal and dual gradient of the cost functions to be optimized, whereas $a_j$ and $b_j$ are, respectively, the coefficient vector and right-hand side value of the $j$th primal constraint $a_j^T x - b_j \geq 0$.

The idea is the following. We start with a dual feasible point $y_0$. From there, we separate the index set $[1:m] := \{1, 2, \ldots, m\}$ into an ordered basic set $\mathcal{B}_k$ with $m_k := |\mathcal{B}_k|$ elements, with $m_k \leq n$ and its complement representing the columns in $\mathcal{N}_k := [1:m] \setminus \mathcal{B}_k$, which can have more than $m - n$ zero elements. Let $A_k \in \mathbb{R}^{n \times m_k}$ and $N_k \in \mathbb{R}^{n \times (m - m_k)}$ be submatrices of $A$ formed by the columns corresponding to $\mathcal{B}_k$ and $\mathcal{N}_k$, respectively. Notice that $\mathrm{rank}(A_k) = m_k$. We will use the notation $\mathcal{R}(A)$ to represent the range of the columns in $A$, while $\mathrm{null}(A)$ denotes its null space, and $|A|$ denotes the number of its nonzeros. Column $j$ of $A$ is denoted by $a_j$. Similarly, $a_i^T$ is row $i$ of matrix $A^T$. Moreover, $b_{\mathcal{B}_k}$ represents an $m_k$ vector with the elements of $\mathcal{B}_k$ of $b$ and $b_{\mathcal{N}_k}$ has the nonbasic elements. The index $k$ will be left out if it is clear from the context, $\| \cdot \|_2^2$ denotes the square of the Euclidean 2-norm, and symbol § will be used in citations as an abbreviation of chapter/section for brevity purposes.

### 2.1. Algorithm

The algorithm works like the simplex method in determining in each iteration a nonbasic variable that enters the basis and a basic variable(s) leaving the basis. The big difference is that it can handle basis deficiency. This means that $m_k \leq n$ and $m_k$ may reduce and grow. A pseudocode is given in Algorithm 1.

- Notice that in line 4, if we have a complementary pair of primal dual feasible points, then $x_k$ is optimal for (P).
- Line 15 determines step size and basis-leaving variable(s), unless in line 12 we found that we have an unbounded dual objective function.

The algorithm does not require a primal feasible starting point. In that sense, it is an exterior method. In earlier studies, we have shown [19] that the algorithm is equivalent to the primal BDA simplex variation given by Pan in [11]. However, we put emphasis on a description, which is easy to understand due to a geometrical interpretation and its independence of implementation details. The algorithm also fits in the primal-feasibility search loop of the Sagitta method described by Santos–Palomo [20], which is related to a loop used in dual active-set methods for quadratic programming. Moreover, Li [21] extended the algorithm to deal with upper and lower bounds on $y$.

Practically, one obtains the primal-feasibility phase I of Dax's 1978 [22] idea for Rosen's 1960 paper [10] when one replaces the min-ratio test by a most-obtuse-angle row rule to determine the leaving variable and removing an objective function consideration. In this

context, we focused earlier [23] on the classical example of Powell [24], which illustrates the cycling behaviour of a most-obtuse-angle simplex pivoting rule described in [12,25].

---

**Algorithm 1** Pseudocode active-set algorithm.

---

1: Set $k \leftarrow 0$, $y_0$ feasible point of (D) with corresponding $\mathcal{B}_0$ and $A_0$
2: $x_k \leftarrow$ solves $A_k^T x = b_{\mathcal{B}_k}$ with residuals $r_k$ following from $r_{\mathcal{N}_k} = N_k^T x_k - b_{\mathcal{N}_k}$
3: **if** $r_k \geq 0$ **then**
4:     **return** $x_k$ is optimal
5: select $p \in \mathcal{N}_k$ with $r_{kp} < 0$
6: **if** $a_p \notin \mathcal{R}(A_k)$ **then**
7:     $\mathcal{B}_{k+1} \leftarrow \mathcal{B}_k \cup \{p\}, \mathcal{N}_{k+1} \leftarrow [1:m] \setminus \mathcal{B}_{k+1}$, set $A_{k+1}$
8:     $y_{k+1} \leftarrow$ follows from $A_{k+1} y_{\mathcal{B}_{k+1}} = c$, $k \leftarrow k + 1$, goto step 2
9: **else**
10:     solve $A_k \delta = a_p$
11:     **if** $\delta \leq 0$ **then**
12:         **return** Unbounded $\mathcal{L}$
13:     **else**
14:         Direction $d_k$ is initially unit vector $e_p$ and elements of $\mathcal{B}_k$ follow from $-\delta$
15:         Step-size $\tau \leftarrow \min_{q \in \mathcal{B}_k} \{ \frac{y_{kq}}{-d_{kq}} | d_{kq} < 0 \}, Q \leftarrow \text{argmin}_{q \in \mathcal{B}_k} \{ \frac{y_{kq}}{-d_{kq}} | d_{kq} < 0 \}$
16:         $y_{k+1} \leftarrow y_k + \tau d_k, \mathcal{B}_{k+1} \leftarrow \mathcal{B}_k \setminus Q \cup \{p\}, \mathcal{N}_{k+1} \leftarrow \mathcal{N}_k \cup Q$
17:         Set $A_{k+1}, k \leftarrow k + 1$, goto step 2

---

The algorithm ends with an optimal solution for $(P)$ and $(D)$. However, the solution does not necessarily correspond to a square basis. Notice that the algorithm maintains dual feasibility and complementary slackness. In fact, two algorithmic strategies now emerge. One of them, Reliable Sagitta, relies on starting Algorithm 1 from a dual feasible point. The other one, Sparse Sagitta—abbreviated as Sagitta—starts Algorithm 1 as soon as a solution of $Ay = c$ is found with a suitable, linearly independent subset of columns of $A$ where $y$ is possibly violating nonnegativity. These two alternatives, along with the chosen touchstone for a fair comparison, are outlined in Section 2.2.

*2.2. Implementation*

To run the algorithm, we first find a feasible dual vertex, i.e., a vector $y \geq 0$ such that $Ay = c$ with columns of $A$ corresponding to positive components of $y$ being linearly independent. This is called a phase-I procedure. One option is to use the procedure of the nonnegative least squares (NNLS) approach described in [19,26]. One can apply the NNLS algorithm [27] or Dax algorithm [28] to the positive semidefinite quadratic problem

$$\min_{y \geq 0} \frac{1}{2} \|Ay - c\|_2^2, \tag{2}$$

which implies solving a sequence of unconstrained least squares problems of the form $\min \|A_k z - c\|_2$ (NNLS) or of the form $\min \|A_k w - s_k\|_2$ (Dax). Such an approach does not require artificial variables and obtains a feasible direction of the feasible region $\mathcal{F}$ (i.e., a so-called direction of $\mathcal{F}$) or a dual basic feasible solution with $m_k \leq n$. It appears that the computation can be described in terms of a primal null-space descent direction [29]. Instead of (2), one can focus on its Wolfe dual, which is a least-distance, strictly convex quadratic problem

$$\min \frac{1}{2} \|u\|_2^2 \quad \text{s.t.} \quad A^T u \geq v, \tag{3}$$

where $v := A^T c$. This also implies solving a sequence of least squares problems. To solve (3), one can use any of the primal, dual, and primal dual active-set methods introduced in [30]. Note that a primal direction $d := u - c$ is easy to generate.

Once a dual feasible vertex $y^*$ is obtained, a phase-II procedure is performed to reach optimality. A traditional way to continue is to artificially enlarge $\mathcal{B}$ from $m_k$ to $n$ columns and then apply the primal simplex method to $(D)$ starting from a degenerate dual vertex of $\mathcal{G}$. However, the interesting approach we follow is not to artificially enlarge $\mathcal{B}$, but to use $y^*$ as the starting point for a nonsimplex active-set method for linear programs in standard form as sketched in Algorithm 1. We shall refer to this first algorithm for solving linear programs, such as Reliable Sagitta (RELSAG v1.1), for which termination is guaranteed under a dual nondegeneracy assumption (see [17]).

The second variant of phase I to solve (2) adapts the inner loop, called Loop B in [27] and step S2 in [29]. This loop sequentially removes indices from $\mathcal{B}$ of those columns of $A_k$ corresponding to negative multiplier least-squares estimates $\bar{y}_{B_i}$ of actual Lagrange multipliers $y_{B_i}$ for $(P)$. We substitute this loop by the following steps to determine the search direction: Compute temporal variables $\bar{y}_B$ solving

$$\min \| A_k y_B - c \|. \tag{4}$$

Then we set $y \leftarrow [\bar{y}_B; 0]$, and take $d \leftarrow A_k \bar{y}_B - c$ and go on to prepare the next iteration. Notice that $\bar{y}_B$ is not constrained to be nonnegative.

This NNLS modification finishes with a dual solution $y^*$ verifying $Ay = c$, but nonnegativity does not hold in general. Hence the primal-feasibility search loop is not ensured to start from a feasible solution $y \in \mathcal{G}$, and termination is not guaranteed. In the event that this loop ends, we can check whether $y \geq 0$ stops with the optimal solution, or deletes a negative element $y_{B_i}$ to restart the whole process. Note that we have delayed the elimination of negative multipliers until they have become actual multipliers. We shall refer to this second algorithm for solving linear programs such as Sparse Sagitta (SPASAG v1.1).

The comparison of the two nonsimplex active-set methods described above was done with the commercial code of Holmström TomLab LPSIMPLEX v3.0 [31] sparse primal simplex implementation, applied to solve $(D)$. Details of this MATLAB solver can be found in http://tomlab.biz/products/base/solvers/lpSimplex.php (accessed on 15 November 2022). This code was previously known as LPSOLVE, but should not be confused with the freely available compiled code *lp_solve* from http://lpsolve.sourceforge.net/5.5/ (accessed on 15 November 2022). To make results comparable, the input of the problems in all the solvers are taken the same way. However, LPSIMPLEX deals with the columns of $A$, whereas RELSAG and SPASAG deal with the rows of $A^T$; hence all of them are under the same conditions when solving $(P)$ and $(D)$.

We also considered comparison with other available software, which appeared less appropriate.

1. LINPROG v2.1 [32] included in MATLAB Optimization Toolbox v2.1. Its active-set method does not allow sparse arithmetic as described above. Other available MATLAB codes like those of Morgan [33] have not been considered because he ultimately rejected the sparse dynamic LU updating that he implemented in favour of a systematic LU restart made from scratch on the basis matrix after each column exchange.

2. We did not compare our code to commercial or freely available compiled codes like MOSEK [34] or MINOS [35], because the running time measurements are not comparable with those of our interpreted MATLAB code. Although solution time is not the only metric that we have used here (number of iterations, quality of solutions), we consider that our solvers must be at least comparable in running time with implementations developed under the same conditions.

3. It is not our aim to use the MATLAB environment to exclude alternative implementations like CPLEX, XPRESS-MP, COIN-OR, or *lp_solve*, to name just a few. We plan to compare against them with larger problems when we develop our own implementation. Moreover, although it is part of the folklore that sparse QR-based methods can suffer more fill-in than similar LU based ones, we expect that easier updating and downdating techniques as those of CHOLMOD described recently by Davis and

Hager in [36] can be used to show the scalability of the developed implementations to larger models.

## 3. Results

This section first describes details of the used settings in Section 3.1 and then focuses on the question of how to make use of specific sparse matrix computations. We discuss results for small, extreme cases from the literature in Section 3.2 to find out which settings are more appropriate for which type of instances. As we will show, this might provide some advantage compared to a general commercial solver like Tomlab. In Section 3.3, we deal with features to handle sparse instances. In earlier work, we also reported on computational tests with 23 out of the smallest 31 NETLIB LP problems, which were originally put forward in standard form ($D$) (i.e., those with no simple bounds on our dual variables), which are currently being analyzed by Im and Wolkowicz [6]. The details of these tests can be found in [37], and a summary of these tests has been included in [17].

### 3.1. Settings

The used implementation (called Sagitta) is intended to find a minimum feasible point of ($P$), without using an initial feasible point. Default values for the parameters are

- maximum number of iterations (default is $5n$);
- set of zero tolerances [38] `tol` (default is $[5 \cdot 10^{-5}, 10^{-8}, 10^{-6}]$); and
- character string named `o` with options. The length of this character string is 5, with `o(1), ..., o(5)` denoting each individual option. Their interpretation has been summarized in Tables 1–3. The setting `o(3)` controls the choice between RELSAG (`o(3)='s'`) and SPASAG (`o(3)='n'`, by default). Reporting of intermediate results and use of updating formulae for residues and search direction are asserted, respectively, with `o(2)='s'` and `o(4)='s'`. The default is 'n' due to lack of a detailed rounding error analysis for such formulae.

**Table 1.** Constraint selection criteria if descent direction $d$ exists.

| o(1) | Meaning |
|------|---------|
| 's' | Most opposite to $d$ no normalization |
| 't' | Most opposite to $d$ with normalization |
| 'a' | Among opposite to $d$, most opposite to $(-c)$ no normalization |
| 'b' | Among opposite to $d$, most opposite to $(-c)$ with normalization |
| 'o' | Among opposite to $d$, most opposite to $d + (-c)$ no normalization |
| 'p' | Among opposite to $d$, most opposite to $d + (-c)$ with normalization |

Our implementation does not make use of simple bounds. Furthermore, it uses a null-space steepest-descent direction $d$ as the search direction. A constraint $a_j^T x \geq b_j$ is said to be opposite to direction $d$ if $a_j^T d < 0$. In the presence of descent direction $d$ (see Table 1), one of the available addition criteria is to add the constraint $a_j^T x \geq b_j$ most opposite with Euclidean normalization to the descent direction if `o(1)='t'` by default. The implementation chooses the opposite $-c$ of the cost vector if `o(1)='b'`, or chooses the difference between the descent direction and the cost vector if `o(1)='p'`. The selection is made among those constraints opposite to $d$ in the last two cases. The implementation can also select not to normalize using the options `o(1)='s'`, `o(1)='a'`, and `o(1)='o'`, respectively. It is worth noting that all criteria given in Table 1 have a strong geometrical interpretation, trying to take advantage of the fact that both $d$ and $-c$ are descent directions.

**Table 2.** Activation criteria in absence of descent direction.

| o(5) | Meaning |
|------|---------|
| 's' | Most violated $a_j^T x \geq b_j$ without normalization (Dantzig) |
| 't' | Most violated with normalization |

**Table 3.** Deactivation criteria.

| o(5) | Meaning |
|------|---------|
| 's' | Most negative multiplier (actual/estimate) without normalization (Dantzig) |
| 't' | Most negative multiplier (actual/estimate) with normalization |

Setting o(5) controls the choice of the addition criterion in the absence of a descent direction (see Table 2). It chooses the most violated constraint with $\ell_2$ norm if o(5)='t' by default or without normalization if o(5)='s'. A constraint deletion (Table 3) can be performed both in the presence or in the absence of a descent direction. The former (only available for RELSAG) chooses the constraint with the most negative multiplier estimate with $\ell_2$ norm if o(5)='t' by default or without normalization if o(5)='s'. The latter (only available for SPASAG) works in a similar way by using actual multipliers instead of multiplier estimates when there is no violated constraint. Moreover, in a constraint exchange this eventuality may be needed in the case of primal feasibility; the constraint to be deleted is chosen in both algorithms by using a min-ratio test similar to that used in the primal simplex method [20,37].

The criteria shown in Table 2 can be thought of as being different ways by which to weigh the residues of the violated constraints. When all of them have unit weight, the classical Dantzig rule applies o(5)='s' and when the weight for constraint $i$ is $\|a_i\|^{-1}$, rule o(5)='t' is selected. In this way, steepest-edge rules work with weight $\|\delta^i\|^{-1} = \frac{1}{\sqrt{1+\|\delta_B^i\|^2}}$, with $\delta_B^i := A_k^\dagger a_i$. Note that, in the simplex case, $a_i \in \mathcal{R}(A_k)$ for all nonbasic $i$ because of the regularity of $A_k$, but the compatibility of $A_k \delta_B^i = a_i$ is not guaranteed in the nonsimplex case, and hence we have to rely on the pseudoinverse $A_k^\dagger := (A_k^T A_k)^{-1} A_k^T$ (see e.g., [39] (p. 17, Equation (1.2.27))). Thus,

$$\|\delta_B^i\| = \|A_k^\dagger a_i\| \leq \|A_k^\dagger\|\|a_i\| \quad \text{implies} \quad \frac{1}{\|\delta_B^i\|} \geq \frac{1}{\|A_k^\dagger\|\|a_i\|}.$$

One can approximate the weights of the steepest edges by

$$\frac{1}{\|\delta^i\|} = \frac{1}{\sqrt{1+\|\delta_B^i\|^2}} \approx \frac{1}{\|\delta_B^i\|} \approx \frac{1}{\|A_k^\dagger\|\|a_i\|}.$$

Notice that the factor $\|A_k^\dagger\|^{-1}$ is shared for all $i$. This reasoning implies that rule o(5)='t' can be thought of as a rough approximate steepest-edge rule. An easier rule to approximate $\frac{1}{\|\delta^i\|}$ in a simplex context has been described by Świętanowski [40].

Both nonsimplex active-set methods have been implemented by using range-space techniques by maintaining the sparse QR factorization of $A_k$. Specifically, the purpose of the modified sparse least squares (MSLS) toolbox v1.3 (that we have developed and described in [29] (§3)) is the sparse updating and downdating of the Cholesky factor $R_k$ of $A_k^T A_k$. The orthonormal factor of the QR factorization of the full column rank matrix $A_k \in \mathbb{R}^{n \times m_k}$ (with $m_k \leq n$) is implicitly maintained as a product $A_k R_k^{-1}$. The triangular factor $R_k$ recurs in sparse form, and is recomputed from scratch by using the sparse MATLAB QR factorization primitive qr when a refactorization is triggered due to an accumulation of rounding errors. Our toolbox consists of the routines SqrIni (initialization of data structures), SqrIns (addition of a constraint to the working set), SqrDel (deletion of a

constraint from the working set), `SqrRei` (refactorization of $A_k$), `Givens` (computation of a Givens rotation) and `EjemMSLS` (application example).

By using MATLAB notation, we store the working set $\mathcal{B}$ in a vector `ACT`. A call to `SqrIni(mode,scal)` is used to initialize the use of NaNs as a lower triangular static data structure `L`, such that `L(ACT,ACT)` is an upper bound for the transposed Cholesky factor of `A(:,ACT)'*A(:,ACT)` independent of what `ACT` is. When `mode='dmperm'`, both the rows and columns of $A$ (and consequently $b$ and $c$) are reordered for $A$ to be in lower block triangular form (LBTF). Hence, the static structure is tightly set up in `L(ACT,ACT)`. With `mode='colmmd'`, only the columns of $A$ (and consequently $b$) are reordered for the static structure set up in `L(ACT,ACT)` to be an upper bound for that Cholesky factor. Scaling techniques can be a priori applied (`scal='noscale'`, by default), like that in which all columns of $A$ and $c$ have a Euclidean norm unit (`scal='txtbook'`), or like that used in CPLEX, where the columns of $A$ are normalized in $\ell_\infty$ norm, and then the same is done with the rows of $A$, when `scal='cplex92'` (see Bixby [41] (p. 271)).

Because we are not using the orthogonal factor due to sparsity considerations, the least squares subproblems are solved by using corrected seminormal equations (CSNE) (see e.g., [39] (p. 126)). Thus, the descent direction $d$ is computed by solving (4) with CSNE. This requires us to perform an iterative refinement step after having found the solution $\bar{y}_B$ of the seminormal equations

$$R_k^T R_k y_B = A_k^T A_k y_B = A_k^T c.$$

Then, $d = A_k \bar{y}_B - c$ is determined as the orthogonal projection of $-c$ onto $\text{null}(A_k^T)$. This means that $d$ is a null-space steepest-descent direction. The same technique is used to solve the compatible system $A_k y_B = c$ when $c \in \mathcal{R}(A_k)$. To test whether $v \in \mathbb{R}^n$ is in $\mathcal{R}(A_k)$, we check whether the residual of $\min \|A_k \delta_B - v\|$ is zero. As a solution $\bar{x}$ of the undetermined system $A_k^T x = b_B$ we take that of the minimum norm by using corrected seminormal equations of the second kind (e.g., [39] (pp. 7, 70, 126)). This requires an iterative refinement step after having found a solution $\bar{z}$ of the seminormal equations of the second kind,

$$R_k^T R_k z = A_k^T (A_k z) = b_B \qquad \text{and} \qquad \bar{x} = A_k \bar{z}.$$

The commercial program TomLab LPSIMPLEX v3.0 [31] enhances a refined MATLAB implementation of the primal simplex method, in which sparse matrices can be handled in the solution process. A sparse QR factorization of the basis matrix is maintained with an explicit orthogonal factor. The updating and downdating is done with customized slight modifications of the dense MATLAB primitives `qrinsert` and `qrdelete`, to facilitate dealing with sparse operands. Notice that the authors of the software claim it handles sparsity exploitation better. Default settings were chosen for LPSIMPLEX.

### 3.2. Extreme Cases

The question we have is what options of Sagitta are necessary to solve extreme cases from the literature. This relates to the general question in the investigation of optimization techniques of which type of algorithms are more appropriate to solve which type (defined by their characteristics) of optimization problems. The investigation does not require big instances, but extremely designed cases like the well-known Klee–Minty box. We report on our finding here.

Gass mentioned in [42] that for any new LP code, it is good to find out which instances may work badly. He specifically designed textbook examples. We gathered 27 from a wealth of different sources in a MATLAB routine called `ExampSag`, which can be found in the Appendix A.

We solved a first set of problems involving primal degenerate (problems 9 and 10) and dual degenerate (problems 14 to 16) problems, corresponding, respectively, to [14] (p. 61), [38] (p. 31), and [43] (p. 136). Sagitta solved all of them without any problem and avoided cycling in the last three cases by using minimum-norm solutions instead of basic

ones when solving the underdetermined compatible systems. It is interesting to note that in the latter two cases, cycling appears if we use basic solutions. This illustrates perfectly that the possibility of a cycle cannot be ruled out for some pivoting rules. We looked for cases unfavourable for the EXPAND technique of solving the degeneration of [44]. We found them in instances of [45] (§2, §4), where the Sagitta code appeared effective in obtaining correct results (problems 19 to 22) for those with a nonzero cost vector.

The literature provides instances that are problematic for interior point methods (Mészáros [46] (pp. 8, 9) and Stojković and Stanimirović [47] (pp. 436, 437)). These are problems 23 to 26 in the Appendix A. Basically, they illustrate that approximate arithmetic may lead to challenges for accepting the optimal solution.

The interesting two cases from Mészáros [46] (pp. 8, 9) could be solved by Sagitta without adjusting the used tolerances. The case where the simplex method was predicted to encounter challenges has a unique solution of $(1, 1, 1, 0)^T$. Tomlab provides as solution $(1, 2, 0, 0)^T$ which implies a row residue $\|c - Ay\|_\infty$ of $10^{-8}$ and Sagitta gives $(1 - 10^{-8}/3, 0, 2 - 10^{-8}/3, 0)^T$ corresponding to a row residue of $\frac{10^{-8}}{3}$.

Consider the two problems presented in Stojković and Stanimirović [47]. The authors claim that by using their method in exact arithmetic they obtain the optimal solutions 0 and $-24{,}196{,}384$ corresponding to active sets $\{1, 3, 12, 16, 17\}$ and $\{1, 3, 6, 12, 14, 16, 17, 20, 21\}$, respectively. Their message is that using interior point methods and approximate arithmetic, the problems cannot be solved. Tomlab, using approximate arithmetic, solves the first of 13 iterations (Iterats) up to $-0.000021164$ with a minimum residual (ResMin) value of 0 and minimum multiplier (MulMin) value $-6.12 \times 10^{-13}$ and is not able to solve the second one.

The results of Sagitta are given in Table 4. For the first problem with label 0, the setting 'annns' provides active set $\{1, 3, 10, 12, 14, 16, 17, 20\}$ with small values for $y_{10}$, $y_{14}$, and $y_{20}$. The setting 'ansns' results in active set $\{1, 2, 3, 10, 12, 16, 17, 20\}$ with small values for $y_2$, $y_{10}$, and $y_{20}$. For the second problem with label 1, Sagitta cannot find the optimal solution with settings 'annns' and 'bnsnt'. Setting 'ansns' gives active set $\{1, 3, 6, 12, 14, 16, 17, 20, 21\}$.

**Table 4.** Settings and results for Stojković and Stanimirović [47] problems.

| N$^o$ | Setting | Optimal Value | Iterats | ResMin | MulMin |
|---|---|---|---|---|---|
| 0 | 'tnnnt' | $-1.725308995636442 \times 10^{-5}$ | 6 | $-3.73 \times 10^{-9}$ | $+5.19 \times 10^{-10}$ |
| 0 | 'snnns' | $+4.457393847569538 \times 10^{-5}$ | 8 | $-7.45 \times 10^{-9}$ | $-1.01 \times 10^{-9}$ |
| 0 | 'bnnnt' | $+7.356378257536402 \times 10^{-5}$ | 9 | $-1.49 \times 10^{-8}$ | $-1.27 \times 10^{-12}$ |
| 0 | 'annns' | $+1.560049165136415 \times 10^{-5}$ | 8 | $-7.45 \times 10^{-9}$ | $-1.01 \times 10^{-9}$ |
| 0 | 'tnsnt' | $-1.725308995636442 \times 10^{-5}$ | 6 | $-3.73 \times 10^{-9}$ | $+5.19 \times 10^{-10}$ |
| 0 | 'snsns' | $-2.117272130221947 \times 10^{-5}$ | 10 | $-2.27 \times 10^{-13}$ | $+1.62 \times 10^{-9}$ |
| 0 | 'bnsnt' | $+7.356378257536402 \times 10^{-5}$ | 11 | $-1.49 \times 10^{-8}$ | $-1.27 \times 10^{-12}$ |
| 0 | 'ansns' | $-2.115917629476000 \times 10^{-5}$ | 17 | $+0.00 \times 10^{0}$ | $-1.27 \times 10^{-12}$ |
| 1 | 'tnnnt' | $-2.419638386181237 \times 10^{7}$ | 9 | $-1.53 \times 10^{-5}$ | $+1.00 \times 10^{0}$ |
| 1 | 'snnns' | $-2.419638389627836 \times 10^{7}$ | 12 | $-1.53 \times 10^{-5}$ | $+4.01 \times 10^{-4}$ |
| 1 | 'bnnnt' | $-2.419638415649983 \times 10^{7}$ | 10 | $-3.82 \times 10^{-6}$ | $+4.44 \times 10^{-4}$ |
| 1 | 'tnsnt' | $-2.419638386181237 \times 10^{7}$ | 9 | $-1.53 \times 10^{-5}$ | $+1.00 \times 10^{0}$ |
| 1 | 'snsns' | $-2.419638389627836 \times 10^{7}$ | 12 | $-1.53 \times 10^{-5}$ | $+4.01 \times 10^{-4}$ |
| 1 | 'ansns' | $-2.419638396563628 \times 10^{7}$ | 12 | $-7.63 \times 10^{-6}$ | $+1.00 \times 10^{0}$ |

We focus now on problems designed by Powell [48] to provide challenges for interior point methods

$$\begin{array}{lll} \min & x_2 & , \ x \in \mathbb{R}^2 \\ \text{s.t.} & \cos(2k\pi/m)x_1 + \sin(2k\pi/m)x_2 \geq -1 & , \ k \in \{1, \ldots, m\}, \end{array}$$

with $m \geq 3$. We focus on the problem for $m \in \{3, 4, \ldots, 500\}$. We run Sagitta with tolerance $\sqrt{\varepsilon}$ and settings $o(1)=$'b' and $o(1)=$'t' by using $o(5)=$'t' in both cases. For the cases where $m$ is a multiple of 4, the algorithm requires only one iteration. For the cases where $m$ is not a multiple of 4, the setting $o(1)=$'b' requires two iterations. This good behaviour

was also predicted by Sherali *et al.* [49] and Künzi and Tzschach [50], Pan [51], as it fits the feasible set. However, the setting $\mathtt{o(1)='t'}$, requires approximately $\log_2(mA + B)$ iterations with $B \approx 10.2$ and $A \approx 0.9$. Notice that this increases to nine iterations for $m = 499$.

Klee and Minty [52] started a worst-case example for the simplex method, where depending on the pivot rule chosen, the number of iterations is exponential in the dimension $n$ of the problem. We focus on the description given by Paparrizos et al. [53] with three variants. For $0 < \varepsilon \leq \frac{1}{3}$, the first formulation of the problem is

$$
\begin{aligned}
\max \quad & \sum_{j=1}^{n} \varepsilon^{n-j} y_j && , y \in \mathbb{R}^n \\
\text{s.t.} \quad & y_1 \leq 1 \\
& 2 \sum_{j=1}^{i-1} \varepsilon^{i-j} y_j + y_i \leq 1 && , i \in \{2, \ldots, n\} \\
& y \geq 0.
\end{aligned}
$$

The second formulation follows from reordering the columns of matrix $A$:

$$
\begin{aligned}
\max \quad & \sum_{j=1}^{n} \varepsilon^{n-j} y_{2j-1} && , y \in \mathbb{R}^n \\
\text{s.t.} \quad & y_1 \leq 1 \\
& 2 \sum_{j=1}^{i-1} \varepsilon^{i-j} y_{2j-1} + y_{2i-1} \leq 1 && , i \in \{2, \ldots, n\} \\
& y \geq 0.
\end{aligned}
$$

The third formulation adds a parameter $\mu := \frac{1}{\varepsilon} \geq 3$:

$$
\begin{aligned}
\max \quad & \sum_{j=1}^{n} \mu^{n-j} y_j && , y \in \mathbb{R}^n \\
\text{s.t.} \quad & 2 \sum_{j=1}^{i-1} \mu^{i-j} y_j + y_i \leq \mu^{2(i-1)} && , i \in \{1, \ldots, n\} \\
& y \geq 0.
\end{aligned}
$$

We introduce a fourth variant (also with parameter $\mu$) in order to obtain the second formulation counterpart with the modified right-hand side. This seems to be a natural generalization, but was not explicitly given by Paparrizos et al. [53]:

$$
\begin{aligned}
\max \quad & \sum_{j=1}^{n} \mu^{n-j} y_{2j-1} && , y \in \mathbb{R}^n \\
\text{s.t.} \quad & 2 \sum_{j=1}^{i-1} \mu^{i-j} y_{2j-1} + y_{2i-1} \leq \mu^{2(i-1)} && , i \in \{1, \ldots, n\} \\
& y \geq 0.
\end{aligned}
$$

In the procedure $\mathtt{KleMinSp}$ in the Appendix A, we extend to the possibility to convert the problem into standard form by adding dual variables in a negative or positive sense; e.g., $\mathtt{KleMinSp(3,10,-3)}$ is the dual of the problem appearing in [43] (p. 270) and in [14] (p. 61). We varied parameter $\mu \in \mathbb{N}$ and activation criteria. The results of the experiments are given in Table 5.

It is clear that without normalisation of the residuals (Dantzig), several instances show an exponential number of iterations. Normalization provides for all combinations a linear number of iterations of an order between $n$ and $2n$. The influence of the position taken by adding the dual slack variables is also interesting. With the setting $\mathtt{o='tnnns'}$ for $\mu \geq 5$, with mode = $-3$, $n + 2^n - 1$ iterations are required. Using mode = 3 only requires $n$. Such differences may disappear with more sophisticated tie-breaking activation criteria. We use for tie-breaking the minimum index criterion (Bland).

Goldfarb [54] designed another deformation of the cube such that steepest-edge rules also require an exponential number of iterations. The parametric problem of Goldfarb [54], with $n \geq 3$, $\beta > 2$ and $\delta > 2\beta$, is given by

$$
\begin{aligned}
\max \quad & (\beta \alpha_n - \alpha_{n-1}) x_{n-1} + \alpha_n x_n && , x \in \mathbb{R}^n \\
\text{s.t.} \quad & 0 \leq x_1 \leq 1 \\
& \beta x_1 \leq x_2 \leq \delta - \beta x_1 \\
& \beta x_j - x_{j-1} \leq x_{j+1} \leq \delta^j - \beta x_j + x_{j-1} && , j \in \{2, \ldots, n-1\},
\end{aligned}
$$

with $\alpha_1 = 1$, $\alpha_2 = \beta$ and $\alpha_{j+1} = \beta\alpha_j - \alpha_{j-1}$, $j \in \{2,\ldots,n-1\}$. The generation of the problem is provided by the procedure `GoldfaSp` in the Appendix A. The results obtained by Sagitta without scaling and with 'dmperm' as ordering method, option setting 'tnnnt', and tolerance $\sqrt{\varepsilon}$ do not provide an exponential growth in the dimension of the number of iterations (see Table 6). The computational time is given in centiseconds (CntSecs), which represent hundredths of a second. This illustrates the requirement of careful numerical procedure and the recoveries.

**Table 5.** Settings for Klee and Minty [52] problems.

| o(5) | o(1) | Mode | Number of Iterations (Iterats) |
|------|------|------|-------------------------------|
| s | t | $\{-3,-4\}$ | $n + \left\{ \begin{array}{ll} 2^n - 1 & \text{, for } \mu \geq 5 \\ 2^{n-1} & \text{, for } (\mu = 3) \vee (\mu = 4) \end{array} \right\}$ |
| s | t | $\{3,4\}$ | $n + \left\{ \begin{array}{ll} 0 & \text{, for } \mu \geq 5 \\ 2^{n-1} - 1 & \text{, for } (\mu = 3) \vee (\mu = 4) \end{array} \right\}$ |
| s | b | $\{-3,-4\}$ | $n + \frac{1}{3}(2^{n+1} - \left\{ \begin{array}{l} 1, \text{ for } n \text{ is odd} \\ 2, \text{ for } n \text{ is even} \end{array} \right\})$ |
| s | b | $\{3,4\}$ | $n + \frac{1}{3}(2^n - \left\{ \begin{array}{l} 1, \text{ for } n \text{ is odd} \\ 2, \text{ for } n \text{ is even} \end{array} \right\})$ |
| t | t | $\{-3,-4\}$ | $n + \left\{ \begin{array}{l} 1, \text{ for } \mu \geq 5 \\ 2, \text{ for } (\mu = 3) \vee (\mu = 4) \end{array} \right\}$ |
| t | t | $\{3,4\}$ | $n + \left\{ \begin{array}{l} 0, \text{ for } \mu \geq 5 \\ 1, \text{ for } (\mu = 3) \vee (\mu = 4) \end{array} \right\}$ |
| t | s | $\{\pm3,\pm4\}$ | $n + 3$ |
| t | {b, a} | $\{-3,-4\}$ | $2n$ |
| t | {b, a} | $\{3,4\}$ | $2n - 1$ |
| t | {t, s, b, a} | $\{\pm1,\pm2\}$ | $nA + B \quad (\varepsilon := \frac{1}{\mu} \to 0 \Rightarrow (A \to 1) \wedge (B \to 0))$ |

**Table 6.** Results for the parametrized Goldfarb [54] problems.

| n | $\beta$ | $\delta$ | Iterats | CntSecs |
|----|---------|----------|---------|---------|
| 3 | 2 | 5 | 5 | 22 |
| 6 | 2 | 9 | 20 | 28 |
| 6 | 3 | 9 | 20 | 33 |
| 6 | 4 | 9 | 20 | 33 |
| 8 | 2 | 10 | 32 | 38 |
| 8 | 3 | 10 | 32 | 38 |
| 10 | 2 | 8 | 46 | 60 |
| 10 | 2 | 10 | 46 | 61 |
| 12 | 2 | 8 | 62 | 82 |
| 12 | 2 | 10 | 62 | 82 |

As a final extreme case, we consider the work of Clausen [55] to design an exponential LP,

$$\begin{aligned} \max \quad & \sum_{j=1}^n \alpha^j y_j & , y \in \mathbb{R}^n \\ \text{s.t.} \quad & 2\sum_{j=1}^{i-1} \beta^{i-j} y_j + y_i \leq \gamma^{i-1} & , i \in \{1,\ldots,n\} \\ & y \geq 0, \end{aligned}$$

with $\alpha = 4/5$, $\beta = 5/4$, and $\gamma = 5$. The idea of the designed problem is to have an exponential number of iterations when using the slack basis for both the simplex method for LPs in standard form and the simplex method for LPs in inequality form applied to the dual of the given problem. The procedure `ClauseSp` in the Appendix A generates the parameters. Let $c$, $A$, and $b$ be the data of the original problem $\max\{b^T y : Ay \leq c, y \geq 0\}$

and let $\bar{c}$, $\bar{A}$, and $\bar{b}$ be the data of the problem $\max\{\bar{b}^T \bar{y} : \bar{A}\bar{y} = \bar{c}, \bar{y} \geq 0\}$ at the output of the routine; then we can select the following variants:

- If `mode=1`, then $\bar{b} = (b^T, 0)^T$, $\bar{A} = [A, I_n]$ and $\bar{c} = c$.
- If `mode=0`, then $\bar{b} = (-c^T, 0)^T$, $\bar{A} = [-A^T, I_n]$ and $\bar{c} = -b$.
- If `mode=2`, then $\bar{b} = (-c^T, 0)^T$, $\bar{A} = [-A^T, I_n]$ and $\bar{c} = b$.
- If `mode=3`, then $\bar{b} = (-b^T, 0)^T$, $\bar{A} = [A, I_n]$ and $\bar{c} = c$.

The variants facilitate applying Sagitta to problem $\min\{\bar{c}^T x : \bar{A}^T x \geq \bar{b}\}$ in unfavourable circumstances for the simplex method for LP in standard form (if `mode=1`) and for the simplex method for LP problems in the inequality form (if `mode=0`). The other modes play the same role with the original problem of minimization instead of maximization, as noticed by Murty [56].

The results obtained by neither scaling nor preordering and with tolerances to $\sqrt{\varepsilon}$ are shown in Table 7 for varying values of `o(5)` and `o(1)`. We use for the number of iterations

$$u_t := 2^{t-1} + t, \qquad \Delta u_t := \frac{1 + ((t+1)\bmod 2)}{3}\left(4^{\lfloor \frac{t-3}{2}\rfloor + 1} - 1\right). \tag{5}$$

The number of iterations for modes 0 and 2 on the one hand, and 1 and 3 on the other hand are very similar, separated by a comma in the last column. The last two experiment rows of the table reflect exponential behaviour. For the first three rows, the behaviour of Sagitta with `o(3)=n` was surprisingly polynomial, requiring at most $n + 2$ iterations by using option setting '`tnnnt`' in each mode.

**Table 7.** Settings for Clausen [55] problems, $u_n$ and $\Delta u_n$ according to (5).

| `o(5)` | `o(1)` | Mode | Number of Iterations (Iterats) |
|---|---|---|---|
| `{s, t}` | `{s, t, a, b}` | $0, 2$ | $n, n$ |
| `t` | `t` | $1, 3$ | $n+1, n+2$ |
| `t` | `{a, b}` | $1, 3$ | $2n-1, 2n$ |
| `s` | `t` | $1, 3$ | $u_n - 1, u_n$ |
| `s` | `{a, b}` | $1, 3$ | $1 + u_{n-1} + \Delta u_{n-1}, u_n + \Delta u_n$ |

### 3.3. Dealing with Sparse Instances

Preliminary experiments were carried out on classical problems with sparse matrices to compare the Sagitta sparse treatment with the one used by TomLab LpSolve v3.0. Initial problems of interest at the time of the appearance of the simplex method [57] (first published in) in 1947 were diet problems, such as the favorite Stigler 9-variable, 77-constraint [58] (p. 3), [59] (pp. 551–567) problem. Instead, we focus on the problems by Andrus and Schäferkotter [60] (p. 582), consisting of

$$\begin{array}{lll} \max & \sum_{i\in\{1}^n (3i - 1)x_i & , x \in \mathbb{R}^n \\ \text{s.t.} & 1 \leq 2x_i + x_{i+1} \leq 2 & , i \in \{1, \ldots, n-1\} \\ & 1 \leq 2x_n \leq 2. \end{array}$$

Procedure `AndSchSp` in the Appendix A generates the problem data given the chosen input parameters. The number of iterations and the computation time (in centiseconds) for different values of $n$ can be found in Table 8. We chose '`dmperm`' as the ordering option as it allows $A$ to be banded, and we did not scale '`noscale`' the problem, using default options for Sagitta with all tolerances to $\sqrt{\varepsilon}$. LpSolve needs to perform $n + 1$ iterations and was unable to solve problems with $n = 1000$ and $n = 2500$ after 9 h of execution due to time limits and memory problems. Sagitta uses typically $n$ iterations and took 20 s compared to almost 6 min that LpSolve needed to solve the problem with $n = 500$. The problems with $n = 1000$ and $n = 2500$ were solved in almost a one and a half minutes and just over 13 min, respectively. Note that Sagitta needed $n$ iterations, whereas TomLab needed $n + 1$.

**Table 8.** Andrus and Schäferkotter [60] diet problems.

| n | **TomLab v3.0** | | **Sagitta v1.1** | |
|---|---|---|---|---|
| | **Iterats** | **CntSecs** | **Iterats** | **CntSecs** |
| 50 | 51 | 258 | 50 | 60 |
| 100 | 101 | 807 | 100 | 126 |
| 200 | 201 | 4048 | 200 | 379 |
| 500 | 501 | 52,531 | 500 | 1972 |
| 1000 | • | • | 1000 | 8634 |
| 2500 | • | • | 2500 | 79,296 |

A widely used class of problems to test the performance of a new LP method is due to Quandt and Kuhn [61]:

$$\begin{aligned} \max \quad & e^T y & , y \in \mathbb{R}^n \\ \text{s.t.} \quad & Qy \leq 10^4 e & , Q \in \mathbb{N}^{n \times n} \\ & y \geq 0, \end{aligned}$$

with $e$ the all-ones vector and $q_{ij} \in \mathbb{N}^+$ randomly chosen in $\{1, 2, \ldots, 10^3\}$. This causes matrix $Q$ to be dense. The procedure `KunQuaSp` in the Appendix A generates the data of the problem from a nonsingular sparse $Q$ of density $10^{-5}$ and condition number 20 by means of `round(1000*sprand(n,n,1e-5,1/20))`, such that $|Q| \approx n$. Running the Sagitta routine, we choose '`colmmd`' as the ordering option and not to scale '`noscale`' the problem and used default option settings. For each value of $n$, 10 problems were solved; the average number of iterations and the average computation time (in hundredths of a second) for different values of $n$ are reported in Table 9. Note that as before, Sagitta requires $n$ iterations whereas TomLab needs $n + 1$.

**Table 9.** Sparse Quandt and Kuhn [61] problems.

| n | **TomLab v3.0** | | **Sagitta v1.1** | |
|---|---|---|---|---|
| | **Iterats** | **CntSecs** | **Iterats** | **CntSecs** |
| 50 | 51 | 215 | 50 | 43 |
| 100 | 101 | 835 | 100 | 86 |
| 250 | 251 | 8199 | 250 | 400 |
| 500 | 501 | 60,136 | 500 | 1531 |
| 1000 | • | • | 1000 | 6660 |
| 2500 | • | • | 2500 | 59,098 |

Chen et al. [62] (§8) presented a new primal-dual method and coded it in MATLAB. To test it, they generated sparse random testing problems where the constraints are tangential to the unit ball. Consider a matrix $A \in \mathbb{R}^{n \times m}$ with Gaussian-distributed elements generated in MATLAB according to `sprandn(n,m,dens,1)`. Consequently, $A$ is a nonsingular, well-conditioned matrix with random normally distributed elements. Moreover, let index number $g$ be uniformly selected from $\{1, 2, \ldots, m\}$. In the Appendix A, procedure `ChPaSaSp` generates a sparse problems of the form

$$\begin{aligned} \max \quad & y_g & , y \in \mathbb{R}^m \\ \text{s.t.} \quad & \sum_{j=1}^m A_{ij} y_j = 1 & , i \in \{1, \ldots, n\} \\ & y \geq 0. \end{aligned}$$

As in Chen et al. [62] (§8), for each of the combinations of interest of number of rows $n$, number of columns, $m$ and density $d$, we generate blocks of 20 problems and average the number of required iterations. Moreover, we also average the elapsed time (in hundredths of a second). In Sagitta, we set '`colmmd`' as the ordering option, and we do not scale '`noscale`' the problem by using default options.

As combinations of interest we first chose the same combinations as Chen et al. [62] §8, and obtained the results shown in Table 10. One can observe that, although Sagitta had a slight advantage in terms of number of iterations, TomLab is clearly faster. This is due to the fact that the problems are not large ($n \leq 150$) and above all due to the fact that the maximum skeleton $L$ is not sparse at all, i.e. $|L|$ is between 75% and 85%.

**Table 10.** High-density sparse problems of Chen et al. [62].

| | | | TOMLAB V3.0 | | SAGITTA V1.1 | |
|---|---|---|---|---|---|---|
| **m** | **n** | **d** | **Iterats** | **CntSecs** | **Iterats** | **CntSecs** |
| 50 | 25 | 0.3 | 39 | 95 | 39 | 117 |
| 100 | 25 | 0.3 | 46 | 110 | 46 | 162 |
| 150 | 25 | 0.3 | 53 | 133 | 50 | 191 |
| 200 | 25 | 0.3 | 54 | 145 | 49 | 196 |
| 100 | 50 | 0.1 | 71 | 271 | 67 | 381 |
| 150 | 50 | 0.1 | 83 | 328 | 79 | 594 |
| 200 | 50 | 0.1 | 92 | 379 | 89 | 774 |
| 150 | 100 | 0.1 | 137 | 1041 | 128 | 2686 |
| 200 | 100 | 0.1 | 158 | 1237 | 153 | 3975 |
| 200 | 150 | 0.1 | 210 | 2673 | 185 | 8547 |

To know the behavior for sparser combinations, i.e., $|L|$ does not exceed 10%, we reran the experiments for such instances. The results are given in Table 11. The results of these sparse experiments show that that Sagitta has a slight advantage in terms of number of iterations, but is also faster. The dimension $n = 250$ is the largest of those solved with TomLab. It took Sagitta 4 s to solve the problem compared to the 83 s required by TomLab. The two largest problems were solved by Sagitta in $n$ iterations requiring 14 and 57 s, respectively.

**Table 11.** Low-density sparse problems of Chen et al. [62].

| | | | TOMLAB V3.0 | | SAGITTA V1.1 | |
|---|---|---|---|---|---|---|
| **m** | **n** | **d** | **Iterats** | **CntSecs** | **Iterats** | **CntSecs** |
| 50 | 25 | 0.06 | 27 | 73 | 27 | 24 |
| 100 | 25 | 0.06 | 30 | 85 | 30 | 40 |
| 150 | 25 | 0.06 | 31 | 92 | 30 | 53 |
| 200 | 25 | 0.06 | 32 | 101 | 31 | 68 |
| 100 | 50 | 0.02 | 51 | 222 | 50 | 37 |
| 150 | 50 | 0.02 | 51 | 233 | 51 | 40 |
| 200 | 50 | 0.02 | 51 | 243 | 51 | 41 |
| 150 | 100 | 0.02 | 103 | 885 | 102 | 288 |
| 200 | 100 | 0.02 | 107 | 938 | 103 | 357 |
| 200 | 150 | 0.02 | 161 | 2339 | 154 | 1865 |
| 500 | 250 | 0.001 | 251 | 8333 | 250 | 400 |
| 750 | 500 | 0.001 | ● | ● | 500 | 1374 |
| 1500 | 1000 | 0.0001 | ● | ● | 1000 | 5706 |

## 4. Discussion

A code has been presented for linear programming to take care of sparsity and basis deficiency. Extreme cases have been taken from the literature, and the performance has been investigated by using a commercial MATLAB solver Tomlab as benchmark. We structured this article as shown in Figure 1 and found that

- the code is numerically robust (cf. Section 3.2), despite the fact that it dispenses with the orthogonal factor of the QR factorisation and in no case uses more than one iterative refinement step;

- the computational effort in choosing suitable pivoting rules is quite encouraging; and
- the code reveals advantages for problems that are really sparse (cf. Section 3.3), both in terms of iterations and computational time, compared to a commercial software with comparable features.

Questions on the simplex method originate from its beginning in 1947. However, there is still a lot of ongoing research; in the last three years one can find many publications, including [1–8], just to cite a few. Although they mainly focus on the simplex method (the second volume of forthcoming renewed edition of Pan's book [8] being the exception), some of the ideas they pointed out could be of interest to be incorporated into a deficient basis framework, thus reducing even more the necessary number of iterations.

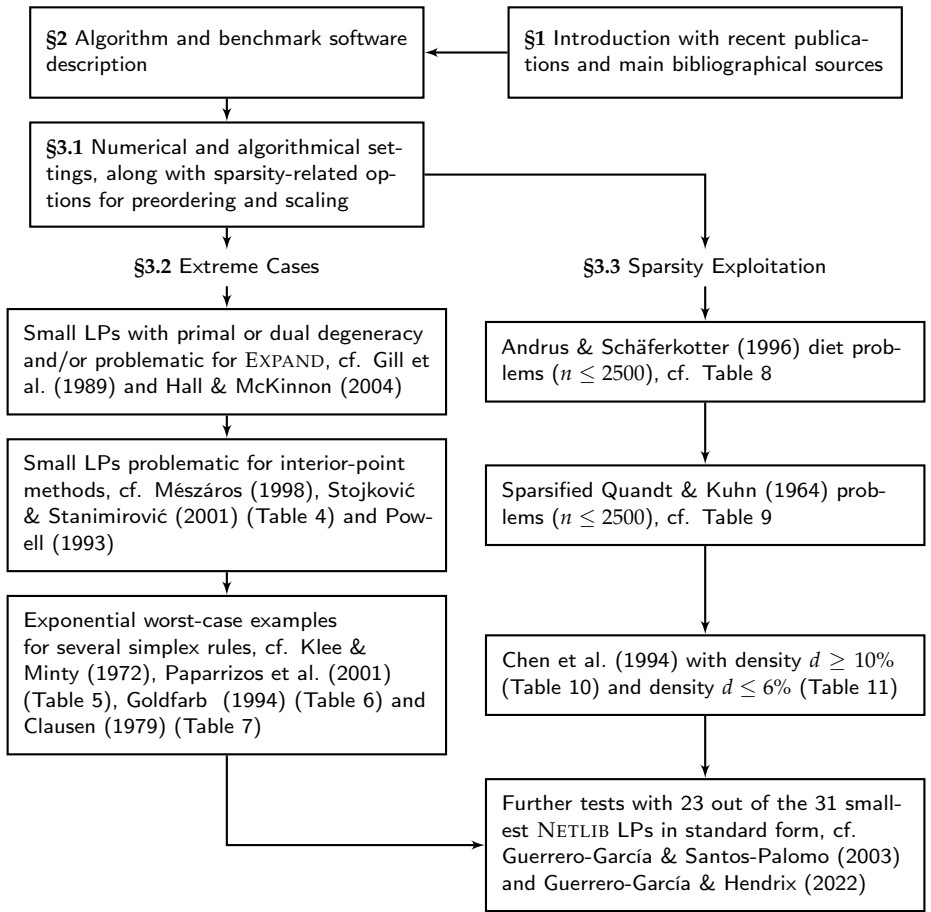

**Figure 1.** Concluding graph of the article, with references Gill *et al.* [44], Hall and McKinnon [45], Mészáros [46], Stojković and Stanimirović [47], Powell [48], Klee and Minty [52], Paparrizos et al. [53], Goldfarb [54], Clausen [55], Andrus and Schäferkotter [60], Quandt and Kuhn [61], Chen et al. [62], Guerrero-García and Santos-Palomo [37] and Guerrero-García and Hendrix [17].

**Author Contributions:** P.G.-G. designed the research, experimental software in MATLAB and designed and analyzed the experimental results. E.M.T.H. discussed results and contributed to the writing of the original draft and polished the final versions of the manuscript. All authors have read and agreed to the published version of the manuscript.

**Funding:** This work has been funded by grant PID2021-123278OB-I00 from the Spanish Ministry of Science and Innovation.

**Data Availability Statement:** All data was presented in the main text.

**Conflicts of Interest:** The authors declare no conflict of interest.

**Abbreviations**

The following abbreviations are used in this manuscript:

| | |
|---|---|
| LP | Linear Programming |
| BDA | Basis-Deficiency-Allowing |
| NSA | Non-Simplex Active-set |
| NNLS | Nonnegative Least Squares |
| MSLS | Modified Sparse Least Squares |
| CSNE | Corrected Semi-Normal Equations |

**Appendix A. MATLAB Code to Reproduce the Used Instances**

```
function [c,A,b,equations]=ExampSag(n)
switch n
case 1  % Example 1, no feasible solution
c=[-1 -3 1]'; b=[-4 4 3 zeros(1,3)]';
A=[-1 -1 -2; -1 0 1; 0 0 1; eye(3)]';
case 2  % Example 2, unbounded solution
c=[1 4 0 -1]'; b=[-2 4 -4 2 0 0 0]';
A=[1 -2 1 -1; 2 1 2 -2; -2 -1 -2 2; 1 0 -3 1; 1 0 0 0; 0 1 0 0; 0 0 0 1]';
case 3  % Example 3, bounded solution
c=[-4 -2 -6]'; b=[-3 -2 -1 0 0]';
A=[-1 0 -1; -1 -1 -1; -1 1 -2; 1 0 0; 0 -1 0]';
case 4  % [Gill, Murray and Wright, 91, p. 381]
c=[1, 10]'; b=[3, 1, -2]';
A=[1 5; 5 -1; -1 -1]';
case 5  %simplicial case [Gill, Murray and Wright, 91, p. 345]
c=[1, 10]'; b=[2.5, -1, 1, -4.7, -10.2]';
A=[4 1; 1 -1; -1 3; -1 -2; -5 -1]';
case 6  % Cycle example with textbook rule [Gill, Murray and Wright, 91, p. 351]
c=[-2, -3, 1, 12]'; b=zeros(6,1);
A=[eye(4); 2 9 -1 -9; -1/3 -1 1/3 2]';
case 7  % Myopic example [Gill, Murray and Wright, 91, p. 397]
c=[-1.2, -1]'; b=[-2.5, -1.4, 0, 0]';
A=[-1.5 -1; -1 0; 1 0; 0 1]';
case 8  % One restriction parallel and the other perpendicular to objective gradient
c=[1, 2]'; b=[4, 8]';
A=[-2 1; 1 2]';
case 9  % Problem KM02
c=[-1.2 -1.0]'; b=[-2.20 -3.25 -2.50 -1.40 0.00 0.00]';
A=[2.2 -1.0; -1.5 -1.2; -1.5 -1.0; -1.0 0.0; 1.0 0.0; 0.0 1.0];
v=[A([5 6],:)\b([5 6]) A([4 6],:)\b([4 6]) ...
A([3 4],:)\b([3 4]) A([1 3],:)\b([1 3]) A([1 5],:)\b([1 5])]';
case 10  % Problem KM03
c=[-1.2 -1.0]'; b=[-2.50 -2.20 -3.25 -2.50 -1.40 0.00 0.00 -2.40]';
A=[1.25 -1.00; 2.20 -1.00; -1.50 -1.20; -1.50 -1.00; ...
-1.00 0.00; 1.00 0.00; 0.00 1.00; -1.20 -1.00];
v=[A([6 7],:)\b([6 7]) A([5 7],:)\b([5 7]) A([4 5],:)\b([4 5]) ...
A([4 8],:)\b([4 8]) A([2 8],:)\b([2 8]) A([2 6],:)\b([2 6])]';
case 11  % Problem LP4
c=[7 20 21 5]'; b=[-5 1 7 2 -2 14 2 -3]';
A=[-5 -3  1  8; -1  4  2  0;  5  3  5  1; -1  4  6  1; ...
5  0  6  1;  4  6  4  2;  0  1  0  0;  0  0  0  1]';
case 12  % [Vanderbei, 97] page 234, exercise 14.2
c=-[7 -3 9 2]'; b=sparse([-1 -1 1 1 zeros(1,4)]');
A=sparse([-1 -1 0 0; 0 0 -1 -1; 1 0 1 0; 0 1 0 1; eye(4)])';
```

```
case 13 %Exponential example [Osborne, 85, p. 61], dual Klee-Minty [Nash and Sofer, 96, p. 270]
c=sparse([1 100 10000]'); b=sparse([zeros(1,3) 100 10 1]');
A=sparse([speye(3); 1 20 200; 0 1 20; 0 0 1])';
case 14  % Example degeneracy [Osborne, 85, p. 76]
A=sparse([1 0 0 1 1 1 1; 0 1 0 1/2 -1/2 -5/2 9; 0 0 1 1/2 -3/2 -1/2 1]);
b=sparse([0 0 0 -1 7 1 2])'; c=sparse([1 0 0])';
case 15  % Example degeneracy [Chvatal, 83, p. 31];
A=sparse([1 0 0 1/2 -11/2 -5/2 9; 0 1 0 1/2 -3/2 -1/2 1; 0 0 1 1 0 0 0]);
b=sparse([0 0 0 10 -57 -9 -24])'; c=sparse([0 0 1])';
case 16  % Example degeneracy [Nash and Sofer, 96, p. 136];
c=sparse([0 0 1]'); b=sparse([zeros(1,3) 3/4 -150 1/50 -6]');
A=sparse([eye(3); 1/4 1/2 0; -60 -90 0; -1/25 -1/50 1; 9 3 0])';
case 17  % [Vanderbei, 97] page 231, exercise 2.11
Aorig=sparse([ 1  1 1  0 0 0 -1 0; ...
-1  0 0  1 1 0  0 0; ...
0 -1 0 -1 0 1  0 0; ...
0  0 1  0 1 1  0 1]);
borig=sparse([-1 -8 -9 -2 -7 -3 0 0]');
corig=sparse([1 0 0 1]');
p=colmmd(Aorig');                        % p = 1  3  2  4
for k=1:size(Aorig,2), normas(k)=norm(Aorig(p,k)); end; normas
for k=1:size(Aorig,2)
priyults(:,k)=[min(find(Aorig(p,k)));max(find(Aorig(p,k)))];
end;
[foo,q]=sort(priyults(1,:));
A=Aorig(p,q); b=borig(q); c=corig(p); spy(A);
case 18  % criss-cross example [Zionts, 69, p. 430]
c=[-2 -4 1 3]'; b=[3 -4 zeros(1,4)]';
A=[-1 -3 1 1; -2 -1 -1 1; eye(4)]';
case 19  % Example 2/6-cycle, for EXPAND [Hall and McKinnon, 04, S2]
% for Dantzig rule unbounded
c=[0 0]'; b=[2.3 2.15 -13.55 -0.4 0 0]';
A=[[0.4 0.2 -1.4 -0.2; -7.8 -1.4 7.8 0.4] eye(2)];
case 20  % Example 2/6-cycle, for EXPAND [Hall and McKinnon, 04, S2]
% for Dantzig rule bounded
c=[0 0 1 1]'; b=[2.3 2.15 -13.55 -0.4 0 0 0 0]';
A=[[0.4 0.2 -1.4 -0.2; -7.8 -1.4 7.8 0.4; 1 0 0 0; 0 1 0 0] eye(4)];
case 21  % Example 2/6-cycle, for EXPAND: [Hall and McKinnon, 04, S4], infeasible
c=[0 0 1]'; b=[2.3 2.15 -13.55 -0.4 0 0 0]';
A=[[0.4 0.2 -1.4 -0.2; -7.8 -1.4 7.8 0.4; 0 -20 156 8] eye(3)];
case 22  % Example 2/6-cycle, for EXPAND: [Hall y McKinnon, 04, S4], bounded
c=[0 0 1 1 1]'; b=[2.3 2.15 -13.55 -0.4 0 0 0 0 0]';
A=[[0.4 0.2 -1.4 -0.2; -7.8 -1.4 7.8 0.4; 0 -20 156 8; ...
1 0 0 0; 0 1 0 0] eye(5)];
case 23  % Example interior point challenge [Meszaros, 98, p. 8]
c=[1 2 3+1e-8]'; b=[0 0 0 -1]';
A=[1 0 0 0; 0 1 1 0; 1 1 1+1e-8 1];
case 24  % Example interior point challenge [Meszaros, 98, p. 9]
c=[1e5 1e-5 2e-5]'; b=[0 0 -1]';
A=[1 0 0; 1 -1 1; 1 -1 2];
case 25  % Example interior point challenge [Stojkovic and Stanimirovic, 01, p. 436]
c=[-9791 9789 -9790 -9790 9791 -9789 9789 -9872 -8790]';
b=sparse([1 2 6 10 11 14 15 17 18 19 20],ones(1,11),...
[-8919 -5981 -9892 -3 -9800 -9989 -993 9978 -9687 -9993 9800],22,1);
```

```
A=[8919 -4788 -2 -9733 -3993 0 -1 -1 -9002 -9789 ...
1 -3 -9971 1 -1 -1 -9978 9687 9993 -1 0 0; ...
-8919 -4790 2 -9733 -3993 -2 1 1 -9002 9789 ...
1 -3 -9971 4902 -1 1 9978 -9687 -9993 -1 0 0; ...
8919 -1 -2 0 0 -2 -1 -1 0 9789 ...
-1 0 0 4901 0 -1 -9978 9687 9993 1 0 0; ...
0 -4788 -2 0 0 0 -1 -1 0 -9789 ...
0 0 0 1 -1 -1 0 0 0 0 0 0; ...
8919 -4789 2 9733 3993 0 1 1 9002 0 ...
0 3 9971 1 -1 1 -9978 9687 9993 0 0 0; ...
-8919 4789 -2 9733 3993 2 -1 -1 9002 0 ...
1 3 9971 -4902 1 -1 9978 -9687 -9993 -1 0 0; ...
0 4788 2 -9733 -3993 -2 1 1 -9002 9789 ...
1 -3 -9971 4900 1 1 0 0 0 -1 0 0];
A=[sparse(A); sparse([1 1 2 2],[1 21 16 22],[-1 1 -1 1])];
case 26  % Example interior point challenge [Stojkovic and Stanimirovic, 01, p. 437]
c=[-9791 9789 -9790 -9790 9791 -9789 9782 -9872 -8790]';
b=sparse([1 2 6 10 11 14 15 17 18 19 20],ones(1,11),...
[-8919 -5981 -9892 -3 -9800 -9989 -993 9978 -9687 -9993 9800],23,1);
A=[8919 -4788 -2 -9733 -3993 0 -1 -1 -9002 -9789 ...
1 -3 -9971 1 -1 -1 -9978 9687 9993 -1 1 0 0; ...
-8919 -4790 2 -9733 -3993 -2 1 1 -9002 9789 ...
1 -3 -9971 4902 -1 1 9978 -9687 -9993 -1 0 0 0; ...
8919 -1 -2 0 0 -2 -1 -1 0 9789 ...
-1 0 0 4901 0 -1 -9978 9687 9993 1 0 0 0; ...
0 -4788 -2 0 0 0 -1 -1 0 -9789 ...
0 0 0 1 -1 -1 0 0 0 0 0 0 0; ...
8919 -4789 2 9733 3993 0 1 1 9002 0 ...
0 3 9971 1 -1 1 -9978 9687 9993 0 0 0 0; ...
-8919 4789 -2 9733 3993 2 -1 -1 9002 0 ...
1 3 9971 -4902 1 -1 9978 -9687 -9993 -1 0 0 0; ...
0 4788 2 -9733 -3993 -2 1 1 -9002 9789 ...
1 -3 -9971 4900 1 1 0 0 0 -1 0 0 0];
A=[sparse(A); sparse([1 1 2 2],[1 22 16 23],[-1 1 -1 1])];
case 27  %Instance with equations [Best and Ritter, 85, p. 231]
K = ones(20,1); F0=sparse(8,1); W=1;
F=sparse([1,3,4,6],1,[W,W,2*W,2*W],8,1); C=sparse(10,8,28);
C(:,1)=sparse([1,3,5,6,7],1,[-1/sqrt(65),7/sqrt(113),1,1/sqrt(65),...
5/sqrt(89)],10,1);
C(:,2)=sparse([1,3,6,7],1,[8/sqrt(65),8/sqrt(113),-8/sqrt(65),...
-8/sqrt(89)],10,1);
C(:,3)=sparse([6,8,10],1,[-1/sqrt(65),5/sqrt(89),1],10,1);
C(:,4)=sparse([6,8],1,[8/sqrt(65),8/sqrt(89)],10,1);
C(:,5)=sparse([7,9,10],1,[-5/sqrt(89),1/sqrt(65),-1],10,1);
C(:,6)=sparse([7,9],1,[8/sqrt(89),8/sqrt(65)],10,1);
C(:,7)=sparse([2,4,5,8,9],1,[-7/sqrt(113),1/sqrt(65),-1,-5/sqrt(89),...
-1/sqrt(65)],10,1);
C(:,8)=sparse([2,4,8,9],1,[8/sqrt(113),8/sqrt(65),-8/sqrt(89),...
-8/sqrt(65)],10,1);
N = sparse(0,0);
for k = 1:10
[p1,m1] = size(N); N = sparse([N zeros(p1,2); zeros(1,m1) [1 -1]]);
end;
c = sparse(11,1,1); A = [-C -N; F' zeros(1,20)]; b = [F0;-K];
```

```
equations = logical(sparse(1:8,1,1,28,1)');
case 28  %Experimental case for testing NNLS
A =[ 1 -2  3  1  0;  0  2  0  0  1; -1  6  5 -3  3; -1 -3 -5  2 -2; ...
-2 -4 -4  2 -5; -3 -1 -2  1  2;  1  0  0  0  0;  0  0  1  0  0; ...
0  0  0  1  0]';
b =[ 2  1 -1 -4 -5 -1 0 0 0]'; c=[ -1 6 6 -3 2]';
case 29  %Experimental case for testing NNLS
A =[ 1 -2  3  1  0;  0  2  0  0  1; -1  6  5 -3  3; -1 -3 -5  2 -2; ...
-2 -4 -4  2 -5; -3 -1 -2  1  2;  1  0  0  0  0;  0  0  1  0  0; ...
-3 -2 -7  1 -5]';
b =[ 2  1 -1 -4 -5 -1 0 0 -7]'; c=[ -1 6 6 -3 2]';
case 30  % Example for IMAJNA
A=[-1 0 0 0 0 -1; 0 1 0 -1 0 0; 1 0 -1 0 0 0 ; 0 1 0 -1 0 1; ...
0 0 0 0 1 1 ; 1 0 0 0 0 2 ; 0 0 -1 -1 0 0; 0 1 0 3 0 0 ]';
b=[-7 -1 3 -1 -1 6 7 -6]'; c=[5 0 -2 1 0 4]';
case 31  % Example for IMAJNA
A=[ 1 0 0 0.5 0 1; 0 1 0 -1 0 0; 1 0 -1 0 0 0 ; 0 1 0 -1 0 1; ...
0 0 0 0 1 1 ; 1 0 0 0 0 2 ; 0 0 -1 -1 0 0; 0 1 0 3 0 0 ]';
b=[-7 -1 3 -1 -1 6 7 -16]'; c=[5 0 -2 1 0 4]';
case 32  % [Chvatal, 1983, pp. 180--181]
b = sparse([139 88 133 137 165 zeros(1,6) zeros(1,15)]');
c = sparse([420 415 355 345 160 95 380 395 270 230 310 420 5200 5200 ...
3600 0]');
filas = [1*ones(1,5) 2*ones(1,5) 3*ones(1,5) ...
4*ones(1,6) 5*ones(1,6) 6*ones(1,6) 7*ones(1,6) 8*ones(1,6) ...
9*ones(1,4) 10*ones(1,5) 11*ones(1,5) 12*ones(1,5) ...
13*ones(1,5) 14*ones(1,5) 15*ones(1,5) 16*ones(1,10)];
columnas = [1:5 1:5 1:5 [1:5 6] [1:5 7] [1:5 8] [1:5 9] [1:5 10] ...
[1:3 11] 1:5 1:5 1:5 [1 4 5 6 7] [1 4 5 8 9] [1 4 5 10 11] 2:11];
valores = [1.4 1.8 1.5 1.4 1.4 9.8 2.4 1.4 1.4 1.4 4.0 0.4 1.4 1.4 1.4 ...
2.8 0.6 1.3 1.4 1.5 5.5 2.2 0.4 1.3 1.5 1.2 5.5 2.2 0.4 1.3 1.3 1.2 5.5 ...
2.2 0.6 1.3 1.3 1.2 5.5 2.6 5.8 1.5 1.5 1.2 5.5 0.6 4.0 1.3 5.5 ...
0.6 1.2 1.3 1.3 2.6 0.6 1.8 1.2 1.2 1.2 0.6 1.8 1.5 1.4 1.4 ...
16 12 35 50 50 20 36 50 50 50 16 12 35 50 50 ...
0.1 0.9 0.8 2.3 -ones(1,6)];
A = sparse(filas,columnas,valores,16,26);
A(1:15,12:26)=speye(15);
case 33  %testing NNLS
c = [0.1 0 0.1 0 -1]'; b=zeros(8,1);
A = [1 -2 1 0 -1 -2 1  0; 0 1 -2 1 0 -1 -2  1; 0 0 1 -2 0 0 -1 -2; ...
0  0 0 1  0  0 0 -1; 0 -1 0 -3 0 -24 -4 -128];
otherwise
disp('unknown instance'); c=[]; A=[]; b=[];
end;

function [c,A,b]=AndSchSp(n)
%  Returns sparse diet problem of size n [Andrus and Schaferkotter, 1996]
%
c = -(2:3:3*n-1)'; b = [ones(n,1); -2*ones(n,1)];
A = sparse([],[],[],n,n); A(n,n)=2;
for i=1:n-1, A(i,i)=2; A(i,i+1)=1; end; A = [A' -A'];

% Testing recipe for TomLab v3.0
clear all; global L ACT NAC A b c;
```

```
% [c,A,b]=AndSchSp(1000);  TomLab cannot solve it!!
[c,A,b]=AndSchSp(25); A=sparse(A); b=sparse(b); c=sparse(c);
Prob = lpAssign(-b,A,c,c,sparse(size(A,2),1)); Prob.SolveAlg = 2;
rkeyk=zeros(size(A,2),1); tic; Result = lpSolve(Prob);
ACT=find(Result.x_k)'; NAC=setdiff(1:size(A,2),ACT); R=qr(A(:,ACT),0);
dkoxk=A(:,ACT)*(R\(R'\b(ACT)));
dkoxk=dkoxk + A(:,ACT)*(R\(R'\(b(ACT)-A(:,ACT)'*dkoxk)));
rkeyk(ACT)=Result.x_k(ACT); rkeyk(NAC)=A(:,NAC)'*dkoxk-b(NAC);
k=Result.Iter; toc, k, full(c'*dkoxk), min(rkeyk(ACT)), min(rkeyk(NAC))

function [c,A,b]=ChPaSaSp(m,n,dens)
%Generates sparse random problems [Chen, Pardalos and Saunders, 94, S6]
%
A = sprandn(n,m,dens,1);
for k = 1:size(A,1)
denom = norm(A(k,:),2);
if denom > eps
A(k,:) = A(k,:)/denom;
else
error('Row of A too small in size');
end;
end;
b = sparse(randi(m),1,1,m,1); c = sparse(n,1); c(:) = 1;

function [c,A,b]=ClauseSp(n,mode)
% To generate sparse problems as those of [Clausen, 1979]
%
if nargin < 2, mode = 0; end;
alfa = 4/5; beta = 5/4; gama = 5; A = speye(n);
for k=1:n-1, A = A + diag(2*beta^k*ones(n-k,1),-k); end;
if rem(mode,2)
b = sparse([alfa .^ (1:n)';zeros(n,1)]);
% Errata in [Murty, 1983, p. 437]: chaning sign b of interest
if mode > 1, b = -b; end;
c = sparse([gama .^ (0:n-1)']); A = [A speye(n)];
else
c = -sparse(alfa .^ (1:n)');
% Errata in [Murty, 1983, p. 437]: changing sign c not interesting
if mode > 1, c = -c; end;
b = -sparse([gama .^ (0:n-1)';zeros(n,1)]);
A = [-A; speye(n)]';
end;

function [c,A,b,s]=GoldfaSp(n,beta,delta,a)
%Generates extreme case of Golfarb with n variables, sparse
%
if nargin < 1
n = 3; beta = 3; delta = 7; a = 0;
elseif nargin < 2
beta = 3; delta = 7; a = 0;
elseif nargin < 3
delta = floor(2*beta+1); a = 0;
elseif nargin < 4
a = 0;
```

```
end;
if (n <= 2) | (beta < 2) | (delta <= 2*beta) | (a-floor(a)~=0) | (a < 0) | (a > 2^n-1)
return;
end;
% As an example, for beta = 3 and delta = 7 we get:
% c=-[0 21 8]; b=[0; 0; 0; -1; -7; -49];
% A=[1 0 0; -3 1 0; 1 -3 1; -1 0 0; -3 -1 0; 1 -3 -1];
c = sparse([],[],[],n,1); A = [speye(n); -speye(n)];
b = sparse([],[],[],2*n,1); b(n+1,1)=-1;
for i=2:n
A(i,i-1)=-beta; A(i+n,i-1)=-beta; b(n+i,1)=b(n+i-1,1)*delta;
if i > 2, A(i,i-2)=1; A(i+n,i-2)=1; end;
end;
a2=zeros(1,n); for i=n:-1:1, a2(i)=rem(a,2); a=floor(a/2); end;
alfa(1)=1; alfa(2)=beta; a2=a2-(~a2);
for i=3:n+1
alfa(i)=beta*alfa(i-1)+a2(i-2)*alfa(i-2);
end;
c(n-1,1)=-alfa(n+1); c(n,1)=a2(n)*alfa(n);
if nargout > 3
for j=0:2^n-1
k=j; k2=zeros(1,n); for i=n:-1:1, k2(i)=rem(k,2); k=floor(k/2); end;
for i=1:n
if rem(sum(k2(i:n)),2)==0, d2(i)=0; else, d2(i)=1; end;
end;
d2=fliplr(d2); s(j+1)=polyval(d2,2);
end;
end;
A=A';

function [c,A,b]=KleMinSp(n,base,mode)
%Generates sparse variants of Klee-Minty, cf. [Floudas and Pardalos, 2001, pp. 193--199]
%
A=speye(n);
if abs(mode)<3
c=sparse(ones(n,1));
else
c=sparse(base.^(2*((1:n)'-1)));
end;
for i=1:n
A(i,1:i-1)=A(i,1:i-1)+2*base.^(i-(1:i-1));
end;
if mode > 0,
b=sparse([base.^(n-(1:n)) zeros(1,n)]'); A=[A speye(n)];
else
b=sparse([zeros(1,n) base.^(n-(1:n))]'); A=[speye(n) A];
end;
if rem(abs(mode),2)==0
p=[1:2:(2*n-1) 2:2:2*n]; A(:,p)=A; b(p)=b;
end;

function [c,A,b]=KuhQuaSp(n,dens)
%Generates Kuhn-Quandt sparse problem [Chvatal, 83, p. 46]
%
```

```
if nargin < 2, dens = 1e-5; end;
N = round(1000*sprand(n,n,dens,1/20));
while sprank(N)~=n, N = round(1000*sprand(n,n,dens,1/20)); end;
A = [N speye(n)]; c = sparse(n,1); c(:) = 1e4; b = sparse(2*n,1); b(1:n) = 1;

function [c,A,b]=Powell(m)
%Returns a dense 2D problem with m constraints, challenge for interior point
%
c = [0 1]'; b = -ones(m,1); A = [cos((1:m)*2*pi/m); sin((1:m)*2*pi/m)];
```

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
