# Peer review of "Experiments with Active-Set LP Algorithms Allowing Basis Deficiency"

_computers, doi:10.3390/computers12010003_

Round 1
Reviewer 1 Report
The necessity of this research and the research gap needs to be elaborated in section 1 (Introduction).
It is required to provide more detail on literature and the advantage and weakness of each previous work.
In section 2. please clearly state each notation b and c, x and y, n and m, and (P) and D represent which characteristics.
In section 2 it is suggested that the authors emphesize that how their idea was generated. Base on which previous researches.
Please add a flowchart in section 2 o f the paper to show the stage of the proposed method.
Providing a flowchart for section 2.2 is strongly suggested to represent the stage of implementation.
Please double check the equation 3. Two 2 are in one column. What does this mean?
In line 107: in all the solvers ate ....
Please check the grammar and writing.
It seems that if content of the paper from line 110 to line 131 move to discussion part would be more effective.
Please rewrite the first paragraph of the section 3. This is not well written.
Line 143 contains errors. Please correct.
Please rewrite sentence "5-character string o with options" appropriately.
"maximum number of iterations; default is 5n" Please support by appropriate reference.
Line 154 "with Euclidean normalization (to the descent direction if ..." needs to be double checked.
Please check line 177 from writing viewpoint.
Please recheck Line 224
Please explain the results of table 4. What are Its, ResMin, MulMin?
Please explain "We introduce a 4th variant with parameter µ"
Why this variant has been proposed?
Please cite "Andrus-Schaferkotter’s diet problems" appropriately in the title of table 8.
Please cite "High density sparse problems of Chen, Pardalos and Saunders" appropriately.
in table 10.
In table 10 what are Iterats and CntSecs?
We decided to repeat the experiment ......
Please check for English writing.
"The results of these sparse experiments show that ..." Please check for English writing.
Please add theoretical suggestions and conclusion.
In line 337 what does cf. S3.2 mean?
Please improve the discussion section with comparison of the strengths and weaknesses of the proposed method with literature.
Please add theoretical implications.
Please add some recent papers (2020 - now) to the literature.
Author Response
Please see the attachment with the point-by-point answer to the reviewer. Changes are shown in blue in the new version of the manuscript PDF, with a list of changes (to be deleted in the final version) just before the Appendix.

Reviewer 2 Report
The authors present a linear programming algorithm that handles sparsity and basis-deficiency. The authors have justified why they do not aim to compare their implementation with a state-of-the-art LP solver, like CLP. They also include a wide variety of models in their computational comparison. However, they have not used any really large instance or well-known benchmark instances, e.g, http://plato.asu.edu/ftp/lp2.html
Author Response

(The authors gave the same response as above.)
